# Transmission of Trading Orders through Communication Line with Relativistic Delay

**Peter B. Lerner** [1,2]

1 Anglo-American University, Letenská 120/5, 118 00 Malá Strana, Czech Republic; pblerner@syr.edu or pbl2@psu.edu
2 Device Consultants, LLC, Woodland Drive, State College, PA 16804, USA

**Abstract:** The notion of "relativistic finance" became ingrained in the public imagination and has been asserted in many mass-media reports. However, despite an observed drive of the most reputable Wall Street firms to establish their servers ever closer to the trading hubs, there is surprisingly little concrete information related to the relativistic delay of the trading orders. There is an underlying assumption that faster electronics are always beneficial to the stability of the network. In this paper, the author proposes a modified M/M/G queue theory to describe the propagation of the trading signal with finite velocity. Based on this theory, we demonstrate that, even if the reaction time of the system is negligible, the propagating signal is distorted by simple acts of trading along the transmission line.

**Keywords:** network dynamics; high-frequency trading; signal processing; relativistic delay; queue theory

## 1. Introduction

High-frequency trading (HFT) became feasible and popular with the advent of cheap, easily relocated computing power and memory. The times when humans could intervene in the execution of most of the trading strategies have long gone. Because of that, a reaction to the changing quotes has to be performed by another computer algorithm, which might react to "stale" prices (Angel and McCabe 2012). The opportunity provided by the modern technology leads to faster and faster trading, until the time of reaction of electronics (now, in the nanosecond range) became small with respect to the relativistic time of propagation of signal between major financial centers (New York–Chicago, 3.8 ms; New York–London, 18.7 ms; New York–Tokyo, 36.2 ms).[1] The characteristic time of the first response to a trading signal is $\tau \approx$ 2–3 ms, which roughly corresponds to the computer messages cycling the circumference of New York City and vicinity with the speed of light (Hasbrouck 2016). Inherent latency of trading quotes is even shorter (Bartlett et al. 2019) (see Table 1, *op. cit.*).

The technical advantage of speed was always exploited by traders. While the case of London's Rothschild receiving detailed information about Napoleon's movements on the continent can be anecdotal, the use of postal pigeons has existed since antiquity (Reuters 2007). In fact, the Rothschilds have organized their own information service across the English Channel since the mid-19th century. The first electromechanical fax communicated stock quotes between Lyon and Paris in the 1860s was installed by a physicist and a priest (Caselli 1865), but it was highly impractical because of contemporary limitations on technology, and it was soon replaced by sending coded messages through telegraph. The arms race for the execution time continues to this day. Order execution became fast—time stamps on the order of minutes and seconds were common in the early 21st century—but now have reached the microsecond range, for which relativistic

---

1 Author's estimate, using geodesic distances.

limitations on signal transmission have become essential. To keep up with the progress, one has to find a method of analysis, which is largely independent of extant technology, i.e., practically, of market latency and trading algorithms.

The analytic framework proposed by the author is based on modified Takács's notion of a waiting time (Takacs 1955). We use M/M/G queue theory to describe the propagation of the trading signal with finite velocity (Riordan 1962). However, unlike a standard queue theory, we introduce a signed measure to describe the propagation of quotes. This change, trivial in our context, makes a big difference if a network has a non-trivial topology, the case currently under investigation. As a result, we obtain, in the simplest case of a one-share quote submission, a system of two integrodifferential equations, which can be solved part analytically and part numerically. This system explicitly contains the speed of signal propagation, as well as the delay time in a trading network.

The notion of "relativistic finance" fascinates the public imagination and has been asserted in many mass-media reports (Lewis 2015). However, despite an observed drive of the most reputable Wall Street firms to establish their servers ever closer to the trading hubs, concrete information related to the relativistic delay of the trading orders is surprisingly scarce.

## 2. Literature Review

Major exchanges responded to the immense acceleration with the timestamps up to a (Wang et al. 2020) microsecond, stamping became the norm, and nanosecond stamping gains ground (Securities and Exchange Commission SEC). Geographically separated markets can observe substantially different best executable prices. This happens when the distance between markets exceeds the relativistic time delay of information: $\Delta x > c \cdot \Delta t$, where $c$ is the speed of light and $\Delta t$ is the difference between when the time order reaches the trading venue and the location of the trader. However, there are no universally accepted financial equations, similar to Black-Scholes, which explicitly include the speed of transmission or trading node delay. One approach to bridge the gap between stochastic processes describing heat (Black-Scholes) equation and special relativity was made in (Angst 2011) and *op. cit*. The question is whether insights borrowed from physics or biology can be fruitful; some of the modern authors think they are (Schinckus 2018; Jovanovic et al. 2019).

The chief difference between the propagation of the physical bodies and the information is that information has no mass. Henceforth, the propagation of a trading order for 1000 shares takes about as much time and computer resources as the propagation of a single-share order.

In response to the price signal, either external or endogenous—being generated by an executing algorithm—the terminal (node) of the network generates an order. This order can be canceled on the way if there is the opposite order of a larger size; otherwise, the imbalance propagates further down the network. Our model is designed to quantify imperfections introduced into quote propagation by the finite speed of light and delay introduced by the reaction of distributed nodes along the path of the signal.

The study of information as opposed to the propagation of physical bodies has a checkered history and was mostly performed in the context of communication networks, beginning with the phone exchanges—the predecessor of a modern World Wide Web. Analysis of connectivity of investors' networks and their comparison with the telephone networks have been recently undertaken by Didier Sornette and collaborators, structured around the Swiss Finance Institute in Geneva (Wang et al. 2020). In particular, the approach being taken by us in the next section—basically, the rate equations—is similar to the rate equations in the paper Patterson et al. (2020). This approach can be supported by the possibility to derive rate equations from the microstructure of the financial markets (Yura et al. 2014, 2015).

### 3. M/M/G Transmission Theory

The state of the server at a time, $t$, at a distance, $x$ from the trading server at $x = 0$ (Figure 1; for brevity, we shall call the receiving point the "node x" or simply "the node") can be expressed through a virtual waiting time as follows:

$$p(t, x) = p(w(t) \leq x) \tag{1}$$

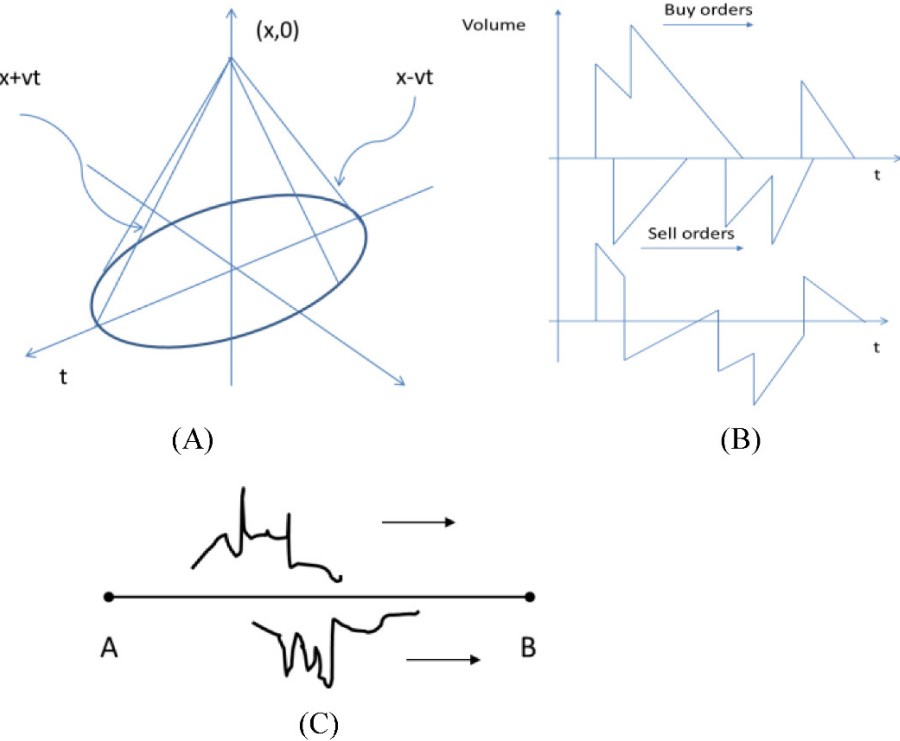

**Figure 1.** (**A**) Relativistic light cone. Only part $x > 0$ is drawn for clarity. (**B**) Schematic shape of the random signal propagating across the line. The upper axis shows "Buy" and "Sell" orders separately, and the lower axis shows the sum of the orders. The vertical axis may correspond to the number of shares traded, or to the aggregate dollar volume, depending on the organization of the trading venue and the type of order. (**C**) "Topology" of the trading network. Exchange is located at point A, and the imbalances are absorbed (cleared) at point B. Traders are distributed along the line AB, according to the Poisson law.

It is equal to the sum of the independent probabilities that the quote does not arrive before the time, $t$, $p_I$, (I = "Idle"), and the probability, $p_r$, that the trading system reacts during the time between $t$ and $t + dt$:

$$p = p_I + p_r \tag{2}$$

The probability $p_I$ can be expressed through $p(t,x)$ as follows:

$$p_I = (1 - a(t)dt) \cdot p(t, x) \tag{3}$$

In Equation (3), $a(t)$ is the arrival rate of the signal or the intensity of the underlying trading process. It is natural to consider this process as an inhomogeneous Poisson or Cox point process, but, for analytic tractability, we shall restrict ourselves to a homogenous

Poisson with intensity $a$ = const below. The probability $p_2$ is expressed through the response function of the trading system $B(x)$:

$$p_r = a(t) \int_0^x B(x-y)d_y p(t,y) \tag{4}$$

The integrodifferential equation of Takács can be written by expanding the probability of the propagation between quote updates[2] at the time $t + dt$ in Taylor series:

$$p(t, x+vdt) = p(t,x) + \frac{\partial p(t,x)}{\partial t} vdt + o(dt) \tag{5}$$

In Equation (5), the parameter $v$ is the velocity of propagation of the electromagnetic signal through the line. Collecting terms from Equations (3)–(5), we obtain the following:

$$\frac{\partial p(t,x)}{\partial t} = v \frac{\partial p(t,x)}{\partial x} - a(t)p + a(t) \int_0^x B(x-y)d_y p(t,y) \tag{6}$$

The difference of our situation with Equation (6) from the original Takács's theory is that we have two probabilities—the probability of the "Buy" signal and the probability of the "Sell" signal. Both equations must look exactly alike, only with their own set of indexes and boundary conditions.

We presume that positive $p_1(t,x)$ (e.g., "buy") and a negative $p_2(t,x)$ (e.g., "sell") signals propagate simultaneously along with the network. Henceforth, we replace Equation (6) with the following system of equations. Technically, this is the system of equations for the signed measures (Cohn 1997), but in what follows, we shall consider probability densities' continuous functions of their argument and interpret derivatives in the sense of distributions:

$$\begin{aligned}
\frac{\partial p_1(t,x)}{\partial t} &= v\frac{\partial p_1(t,x)}{\partial x} - a(t)p_1 + a(t)\int_0^x B_{11}(x-y)d_y p_1(t,y) + a(t)\int_0^x B_{12}(x-y)d_y p_2(t,y) \\
\frac{\partial p_2(t,x)}{\partial t} &= v\frac{\partial p_2(t,x)}{\partial x} - a(t)p_2 + a(t)\int_0^x B_{21}(x-y)d_y p_1(t,y) + a(t)\int_0^x B_{22}(x-y)d_y p_2(t,y)
\end{aligned} \tag{7}$$

In the system of Equation (7), the coefficients $B_{11}(x)$ and $B_{22}(x)$ refer to the modification of buy and sell signals during propagation, and $B_{12}(x)$, $B_{21}(x)$ is the cancellation of signals by the signal of opposite sign.

The system of Equation (7) is linear. Propagation of a quote for one share or 1000 shares takes the same time. However, the next level of realism would include a margin for the short selling, which can be modeled as a reflecting boundary at some critical value of $p_2(t, y_0)$. This problem is investigated elsewhere.

At the next stage of simplification, we assume that the Hawkes-type process governing Equation (7) is, in fact, Poisson (i.e., $a(t) = a = const$). This assumption is not essential for numerical analysis of a network and is provided here as an analytical illustration. We apply the partial Laplace–Stieltjes transform in x-variable, as follows:

$$\varphi_{1,2}(t,s) = \int_0^\infty e^{-sx} dp_{1,2}(t,x)$$

We can, like Riordan (1962), obtain a system of ordinary differential equations and integrate it numerically, taking the initial conditions into account. For elucidation of the analytical behavior of the solutions, we apply Laplace transform in both $t$ and $x$:

$$\varphi_{1,2}(\tau,s) = \iint\limits_0^{+\infty} e^{-sx-\tau t} dp_{1,2}(t,x)$$

---

2   Note that Equation (5) is for free propagation, and it does not contain Ito terms (Jeanblanc et al. 2003). The Ito equation would appear if one considers a slowly varying response in Equation (6), where the last term can be expanded in a Taylor series, as well.

The system of integrodifferential Equation (7) requires initial and boundary condition for each variable:

$$\left\{ \begin{array}{l} p_1(x,0) = f_1(x,0) \\ p_1(x,0) = f_1(x,0) \\ p_1(0,t) = w_1(t) \\ p_2(0,t) = w_2(t) \end{array} \right\} \tag{8}$$

In real life, the boundary conditions of Equation (8) are stochastic, but, in the next equation, we shall consider them arbitrary but known functions.

The system Equation (7) acquires the following form:

$$\begin{pmatrix} \tau - s + a - a\beta_{11}(s) & -a\beta_{12}(s) \\ -a\beta_{21}(s) & \tau - s + a - a\beta_{22}(s) \end{pmatrix} \begin{pmatrix} \varphi_1 \\ \varphi_2 \end{pmatrix} = \begin{pmatrix} \varphi_1(s,0) - sw_1^*(\tau) \\ \varphi_2(s,0) - sw_2^*(\tau) \end{pmatrix} \tag{9}$$

For simplicity and clarity of the analytical solution, we suppose that $\beta_{11} = \beta_{22} = \beta_1(s)$ and $\beta_{12} = \beta_{21} = \beta_2(s)$. This means, quite intuitively, that the transmission coefficients for the buy and sell orders are the same. In the system of Equation (9), $w^*_{1,2}(\tau)$ are functions that are chosen in such a way that the zeroes of numerator and denominator coincide in the region Re($s$) > 0 for Re($\tau$) > 0. Beneš (1957) showed the existence and uniqueness of this choice for Equation (6), but we suggest the existence—but not necessarily uniqueness—as a hypothesis for the system of Equation (9).

The transition matrix in Equation (9) is invertible, except for the curves $\tau = \tau(s)$ in Fourier space, outlined by the zeroes of its determinant:

$$P(s,\tau) = [s - \tau - a(1 - \beta_1(s) - \beta_2(s))][s - \tau - a(1 - \beta_1(s) + \beta_2(s))] \tag{10}$$

The zeroes of the determinant of Equation (10) describe, in effect, conditions for the market clearing of the speed-of-light limited trading system. The curves defined by the equation

$$P(s, \tau(s)) = 0$$

represent the boundaries for the market clearing, which can be interpreted in the language of signal processing. For the waves with the wavenumber $k > s$, the transmission (trading) system acts as a low-pass filter with characteristic frequency $\tau(s)$. For the waves with the wavenumber $k < s$, the transmission system acts as a high-pass filter with the same characteristic frequency. These considerations are presented in the next section, in more physical terms of "sending node" and "receiving node", and "past" and "future" (see Figure 2).

The system Equation (10) has quite a general character. To make it analytically tractable, we assume that the delay by the node is exponentially distributed. This intuitively corresponds to the fact that traders' actions are being governed by the Poisson random process along the length of the communication line:

$$\begin{array}{c} B_{11,12,21,22}(x) \propto e^{-\lambda|x|} \\ \beta_{1,2}(s) \propto \frac{\tilde{\beta}_{1,2}}{s+\lambda} \end{array} \tag{11}$$

where $\tilde{\beta}_{1,2}$ are simply c-numbers. Then, one can reduce zeroes of Equation (10) to the zeroes of the characteristic polynomial:

$$P^*(s,\tau) = (s + \lambda)^2 [s - \tau - a(1 - \beta_1(s) - \beta_2(s))][s - \tau - a(1 - \beta_1(s) + \beta_2(s))] \tag{12}$$

The impulse response function (the Laplace transform of the Green function) for Equation (10) can be found by an inverse Laplace transform. The Laplace original has the form:

$$\hat{P}^{-1}(s,\tau) = (P^*)^{-1}(s + \lambda)^2 \hat{M}(s) \tag{13}$$

In Equation (13), $2 \times 2$ matrix $\hat{M}$ does not have singularities outside of the singularities of its denominator (Riordan 1962; Takacs 1955).

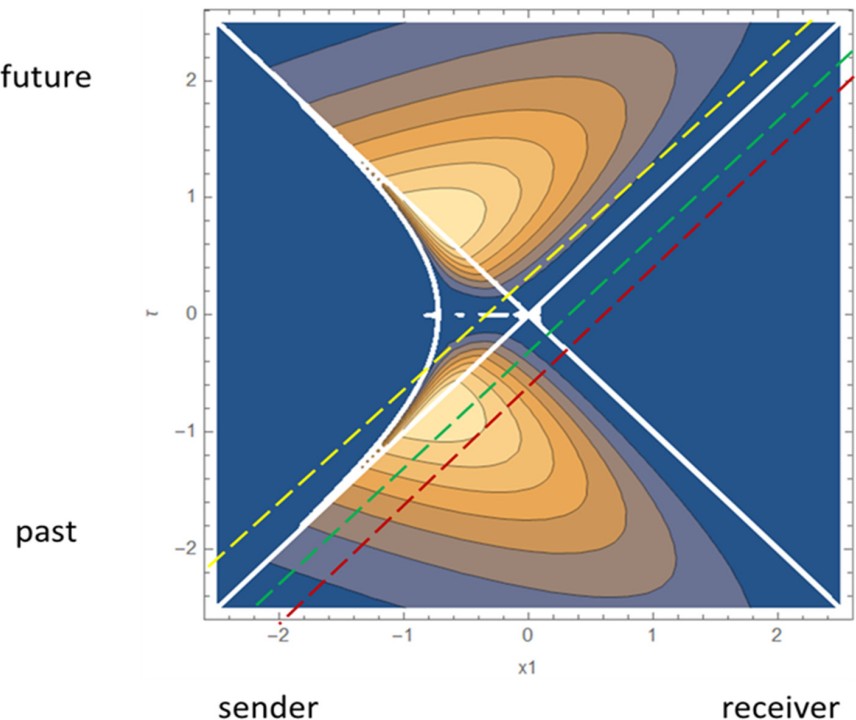

**Figure 2.** Space/time propagation of the signal in the trading system. The signal propagates from the lower left corner of the frame to the upper right corner parallel to the diagonal. The purple line indicates $\tau = x/c - 0.5a$ (the signal is in the future with respect to an observer at the origin), the green line is $\tau = x/c - 0.25a$ (signal approaches the origin), and the yellow line is $\tau = x/c + 0.25a$ (the signal passes the origin). One might consider $a$ as a crude measure of the reaction time of the trading system. Values for the parameters of the Equations (9), (11), and (12) used for all figures are $\lambda = 1.5a$, $\tilde{\beta}_{11} = \tilde{\beta}_{22} = 0.6a$, and $\beta_{12} = \beta_{21} = 0.3a$ and were chosen for the best visual appearance of plots. Hyperbolic structure in the left quadrant indicates the space/time region where the signal is indistinguishable from the noise.

With the approximation of Equation (11), the inverse of the response function is as follows:

$$\hat{K}(x,t) \propto \iint_{C_1,C_2} e^{sx+\tau t} \hat{P}^{-1}(s,\tau) ds d\tau \tag{14}$$

where $C_{1,2}$ are the contours involved in the inverse Laplace transform (Jeanblanc et al. 2003; Davies 2002). The reaction of the system on an arbitrary trading signal $\vec{\Phi} = \begin{pmatrix} \varphi_1(x) \\ \varphi_2(x) \end{pmatrix}$ can then be computed as the convolution of the matrix defined by Equation (12) as follows:

$$\vec{F}(x,t) = \int_0^\infty \hat{K}(x-x',t)\vec{\Phi}(x',0)dx' \tag{15}$$

We do not know anything about the trading signals, which change every millisecond, but in practical applications, one can compute the correlation functions of an arbitrary signal, using Equation (14). The time response to any signal, however, is determined by the kernel, $\hat{K}$.

## 4. Representation of the Propagating Signal

After the simplifications outlined in Section 2, an impulse response (Green) function can be computed semi-analytically. There are several complications involved in the visualization of the results.

First, to visualize the Green function, we must accept a convention that is somewhat different from the one familiar from elementary physics textbooks. In most noncosmological physics contexts, it is natural to imagine a source located at $x = 0$, $t = 0$, and the signal propagating into the future. In the case of trading quotes, we imagine a signal propagating from a distant (but not infinitely distant) past, to an observer in the present (see Figure 1). Indeed, any finite-energy signal propagating from $x = -\infty$ in an absorbing medium will have zero amplitude at the point of origin. Thus, we must impose some finite boundary conditions on the propagating wave.

The second and more important complication is that numerical inverse Laplace transform pays no heed to the need to change the path of integration when the argument of the Laplace integrand crosses the Stokes line (Berry 1988; Meyer 1989). This feature has to be entered by hand into a Mathematica© plot. Intuitively, it corresponds to the situation when the signal is completely absorbed at $x = 0$ and then reemitted by a secondary source in both forward and backward directions (Figure 2).

In Figure 3, we plot the shape of the transfer function near the origin. We observe, naturally, that there is no signal at $x = 0$ before the origin falls into a future cone for the incoming signal. The propagating signal is asymmetric with respect to the back and front wings and undergoes significant distortion when it penetrates beyond an observation point.

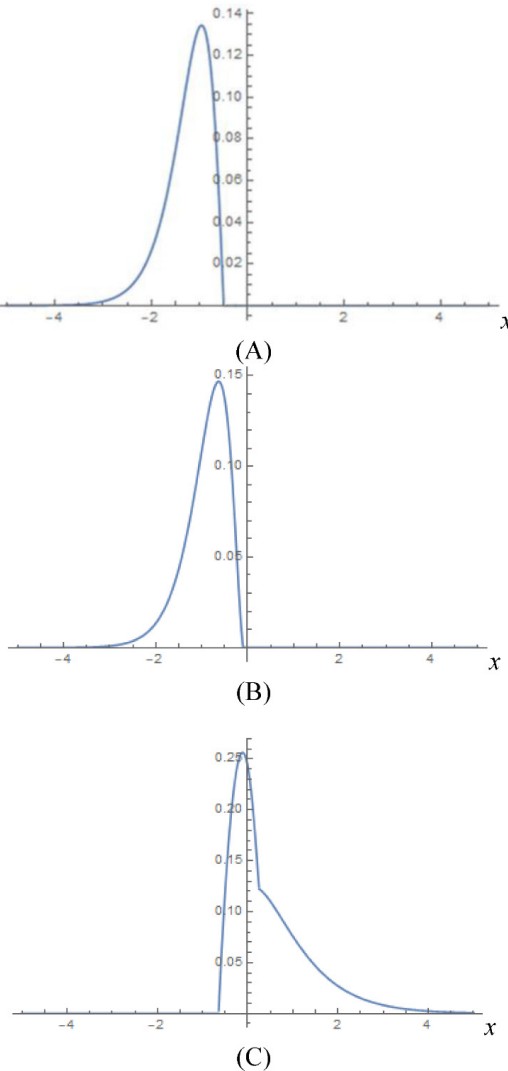

(A)

(B)

(C)

**Figure 3.** The shape of the pulse in arbitrary units (a response to the delta function) in the cases (**A**) $\tau = x/c - 0.5a$, (**B**) $\tau = x/c - 0.25a$, and (**C**) $\tau = x/c + 0.25a$. The spatial coordinate is measured in the units $c \cdot a$ (see Equation (5)).

From our simulations, we can infer that the maximum distortion happens close to the cusp of a light cone and depends on the bandwidth of the correlation signal only weakly. This happens because, close to the cusp of the cone, there is no physical difference between forward- and back-propagating wave packets. Only after some time, related to the time of the network reaction (β-coefficients above), there is a distinct separation between transmitted and reflected signals. This conclusion suggests that, in the building of the remote trading system, one should not strive for the fastest possible reaction time but select the reaction time of the system, which significantly exceeds the additional latency of the trading signal caused by a finite velocity of light propagation. Otherwise, a response to a delayed signal will be close to random.

We have to establish a specific criterion for the fidelity of the transmission of a signal. Because both the baseline distribution and the distribution of an actual transmitted signal are non-stationary, the choice of fidelity measure is far from unique. Fidelity can refer to an initial waveform, as well as to the response of the transmitting line to a delta-shaped distribution. We plot autocorrelation of the Green function as a function of time delay in Figure 4 and Kullback–Leibler distance, $D_{KL}$, between propagating waveforms, interpreted as probability distributions that are dependent on the time delay at the origin (Kullback and Leibler 1951; Burnham and Anderson 2002). In our case, we express delay-dependent Kullback–Leibler distance as follows:

$$D_{KL}(\tau) = \int_{-\infty}^{\infty} K\left(x, \frac{x}{c} - \tau\right) \cdot \log\left(\frac{K\left(x, \frac{x}{c} - \tau\right)}{K\left(x, \frac{x}{c}\right)}\right) dx \qquad (16)$$

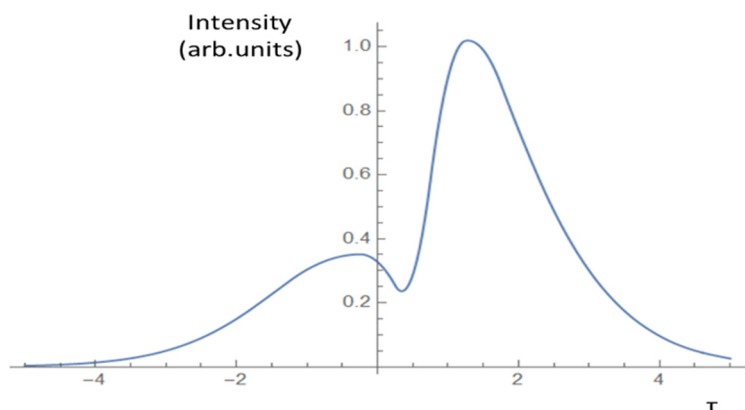

**Figure 4.** Autocorrelation of the square of the Green function kernel $I(\tau) = \int_{-10}^{10} K^2(x, \frac{x}{v} - \tau)dx$ (see Equation (14) in the text), as a function of time delay and $v = 0.75c$. Limits of integration are arbitrary as a truncation to approximate $[-\infty, +\infty]$ with the necessary accuracy.

From Figure 5, we observe that signal with "maximum information" appears at delay $\tau \approx 2$ in the units of *a* in Equations (9)–(11) and not at $\tau \approx 0$, as one can imagine. This consideration has to be taken into account in the construction of HFT systems, of course, based on much more realistic and network-specific models.

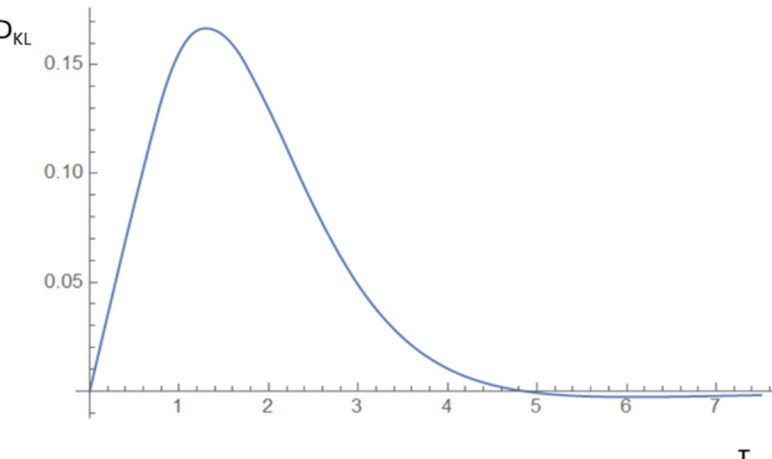

**Figure 5.** Kullback–Leibler distance (Equation (15)) between Green functions kernels as a function of time delay and $v = 0.75c$. A small negative tail is spurious and is related to numerical approximations.

## 5. Discussion

Co-location of the trading servers at exchanges grew from a vogue into imperative during the 2010s. Increasing demands on the trading systems, due to shortening latency, were already recognized in 2011 (Haldane 2011). Practitioners recognized this problem before academics. Of academic work, I can mention the above-cited (Angel and McCabe 2012, 2015) mostly dedicated to regulatory activities, as well as the recent work of the Japanese SPARX group (Yagi et al. 2020). In response to these problems, New York Stock Exchange/Intercontinental Exchange (NYSE/ICE) organized a special division, NYSE Technology, and produced a set of guidelines on the spread of trading venues and limitations to the activities trading firms can undertake at these venues separate for the US and European trading areas (NYSE 2020). However, these regulations deal mostly with trading practices and do not mention technological parameters of the network, except energy and cooling requirements.

In NYSE/ICE documents, it is presumed that each trading firm can organize its Meet-Me-Room (MMR) optimally, without interfering with other trading participants or stability of the network. It is implicitly presumed that telecom providers approved by the exchange can assure harmonization of networks on their own. However, in the open literature (many details of such systems are proprietary), there is a scarcity of information on how transition processes in such systems are quantified and assessed. The author views the contribution of this paper as establishing a connection between informational and physical characteristics of the trading systems.

It is premature to suggest improvements to the trading networks, based on this oversimplified model (see below). However, based on this study, one can propose that the regulatory agencies (FINRA, CFTC, and the like) undertake a study to determine whether the current architecture of trading systems, because of fundamental limitations imposed by special relativity, discriminates against some users or provides them with incentives to manipulate the market, for instance, by stuffing the quotes along the way.

## 6. Conclusions

In the first decade of the 21 century, the transmission rate of trading networks—latency in the parlance of high-frequency traders—began to experience limitations related to the finite limiting speed of the propagation of information (speed of light) (Bartlett et al. 2019). The extremely short latency of modern trading networks created a real possibility that distant traders do not respond to the actual prices on the exchange but to their retarded quotes. The assumption that a faster reaction by the system's electronics is always beneficial for the performance of the network was, to the author's knowledge, never explored in the

literature. This paper proposes an analytical method for how construction of these systems and their stability, during extreme loads, might be analyzed quantitatively.

This paper outlines a theory of signal propagation in the trading network, modeled as an M/M/G queuing network. Namely, the M/M/G station at $x = 0$ services a signal propagating from the left (theoretically, from $-\infty$). A trading signal is modeled as a signed probability measure according to the specification developed in a pioneering work of (Takacs 1955). The update of quotes is represented by the exponentially correlated distribution or a continuous version of the Poisson process.

The main conclusion of this theory is that the response of a trading node instead of being as quick as possible must be harmonized with the parameters of the entire network. The intuitive reason for that is that every trading network is dispersive. Even a white (delta-correlated) noise acquires finite correlation length during propagation in the network. If the current node is located at the apex of the light cone, there is no physical difference between the front of an advanced signal and the tail of a retarded signal. Only when advanced and retarded noises are sufficiently separated in time, can the trading signal be resolved.

In this paper, we studied only the simplest topology of the network—namely, a closed interval with a Poisson-distributed set of responders. Generalization of this theory to the networks with nontrivial topology—basically, weighted planar graphs, requires a different approach, which we are currently investigating.

**Funding:** The author performed this work without any external funding.

**Acknowledgments:** The author wants to thank three anonymous referees for the thoughtful comments and references, which contributed to the significant improvement of this paper.

**Conflicts of Interest:** The author declares no conflict of interest (participation in HFT activities, etc.).

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
