# Peer review of "Transmission of Trading Orders through Communication Line with Relativistic Delay"

_ijfs, doi:10.3390/ijfs9010012_

Round 1

Reviewer 1 Report

Dear author, despite the fact, the paper is devoted to an interesting and topical issue (is logical and coherent), it does not meet the standards of a scientific paper. 

I hope, that the authors will find these comments useful in order to improve the manuscript. Some important comments are listed below.

First of all, the abstract should be reorganized as it does not contain relevant findings. There is a lack of information on new insights and the contribution of the paper. Please, add your crucial findings, main aim and methods used. 

Paper has a short literature review in the Introduction section - this there is neither the introduction nor the literature review. Please, write the literature review as a special section (for example 2. Literature review) and add more up to date literature to underline the importance and topicality of the issue (use the references of last 2-3 years). 

The methodology section (chapter 2 is clear and detailed), please check, if all variables and equations are properly identified and explained. 

In the practical section, the own contribution and the originality of the research should be portrayed and also do not forget to highlight the scientific value of the research.

The discussion section is not written, please add. It could maybe include some implication for the regulation or development of the sector (based on your findings) or some comparative analyses.

The conclusion section is missing practical implication, limitations of the study and well as its further research direction.  

Author Response

Dear Anonymous Reviewer,

Thank you for the thoughtful review of my paper. Below I outline the changes made to comply with your suggestions:

  1. Abstract has been rewritten to highlight the findings of the paper.
  2. The Section 2 "Literature Review" has been added.
  3. I eliminated the confusion of indexes in former Section 2, now Section 3.
  4. I did not add the discussion section, but significantly enlarged the Conclusion section as to include implications for the practitioners and regulators.

Please see the attachment for point to point answer.

Thank you again, for your review. 

Reviewer 2 Report

The abstract demands substantial revision as it fails to accurately summarize the contents of the article. It is necessary to rewrite the article in a more reader-friendly way so that readers know where they are and where they are going. The introduction is way too general while simultaneously insisting on details. The research idea is not properly contextualised, as there is a need of offering a detailed review of relevant literature that help the authors developing the key arguments that support their proposed research. The main contribution should be emphasised more and the concluding statements should be stronger. Lack of high-quality recent peer-reviewed sources to support key points made by the article represents the key area for the development of the article as it stands, and addressing this will constitute most of the work of revision. If the authors rearrange and adopt a critical point of view when writing the theoretical framework, information will be then meaningful. The manuscript requires major revisions to contexualize the merits of the study and potential uses of its methodology in future studies. The analysis strikes me as requiring a bit more depth and to clearly state the contribution that it makes, even though it aims to provide an overview of the topic. Please provide more details regarding the study limitations and strengths and what this means for the study findings.

The cited literature is mostly extremely old. Here are some recent research suggestions that complement your approach (I am not the (co-)author of any of them):

Bin, L., Chen, J., and Ngo, A. X. (2020). “Revisiting Executive Pay, Firm Performance, and Corporate Governance in China,” Economics, Management, and Financial Markets 15(1): 9–32. doi:10.22381/EMFM15120201

Bin, L., Chen, J., and Tran, D. S. (2019). “Exploring the Determinants of Working Capital Management: Evidence across East Asian Emerging Markets,” Economics, Management, and Financial Markets 14(2): 11–45. doi:10.22381/EMFM14220191

Ionescu, L. (2020). “The Economics of the Carbon Tax: Environmental Performance, Sustainable Energy, and Green Financial Behavior,” Geopolitics, History, and International Relations 12(1): 101–107. doi:10.22381/GHIR121202010

Blake, D. (2020). “How Bright Are the Prospects for UK Trade and Prosperity Post-Brexit?,” Journal of Self-Governance and Management Economics 8(1): 7–99. doi:10.22381/JSME8120201

Garcia-Duran, P., and Eliasson, L. J. (2018), “Squaring the Circle: Assessing whether the European Union’s Pursuit of Bilateral Trade Agreements Is Compatible with Promoting Multilateralism,” Journal of Self-Governance and Management Economics 6(1): 7–32. doi:10.22381/JSME6120181

Schinckus, C. (2018). “From DNA to Economics: Analogy in Econobiology,” Review of Contemporary Philosophy 17: 31–42. doi:10.22381/RCP1720183

Author Response

Dear Anonymous Reviewer,

Thank you for the thoughtful review of my paper. Below I outline the changes made to comply with your suggestions:

  1. Abstract has been rewritten to highlight the findings of the paper.
  2. The Section 2 "Literature Review" has been added. In it, I added more recent publications on the subject. I used the list supplied by the referee. First, I added reference to C. Schinkus (last in your list) and the newer publication from the same group. Second, I used bibliography of the papers 1) and 2) in your list, to add publications from D. Sornette and his collaborators. 
  3. I added a paragraph on the study's limitations. Its insights were recapitulated in a significantly enlarged Conclusion section. 
  4. Section "Funding and Acknowledgements" has been added. 

Thank you again for the review. 

Reviewer 3 Report

In the Abstract are explained poor the basics for the paper content. Here is necessary to include also some article findings!

In the Introduction the author mentioned: "relativistic time of propagation of signal between major financial centers....'' without mentioning the source!

Also in the Introduction, the citation mode is improperly designed(please not include the _ symbol before the parentheses, in most cases). This is partial valid for the Section 2, as well.

Even you consider that you need to customize the main paper sections you have to follow the recommendations/instructions for authors (e.g. Sections 2 could be entitled: Materials and Methods and so on). 

Since you numbered the relationships in order to refer to them within the paper you have to revise all the equations in order to be numbered as well (see lines 100, 104, 128, 148). 

You use on the lines 108 and 109 a statement: "...Equations (8) are stochastic, but for now, we shall consider them arbitrary but known functions". Please offer explanations and arguments! Why so?

In the same time, are necessary details related to the Figure 1. B. The schematic shape is based on a particular case or is a general view. The volume is related to a company, a corporation etc. and refers to a numbers of orders or it is about volumes of money traded.

On the other hand I would like the author to focus on Fig.2 and give some other explanations on how this picture could suffer changes, what is important in terms of interpretations etc.

For sure, the Conclusion section needs to be extended in terms of practical use for readers interested in the subject. Also, maybe here or in a separated section (e.g. Discussion) is useful to be mentioned the limits of the approach designed by the author. In the same time, as a key element in the final is mandatory to underline the main author contribution to the field in terms of the paper findings.

For sure the paper could be improved in a proper way and become a useful contribution to the field of transmission of trading orders through communication line by follow and treat all the observations and comments.

Last but not least, the author must include at the end of the paper things like: Funding and Acknowledgments. Please pay attention to the References Section (please remove the _ symbol placed before the names in the list starting with the B letter).

Author Response

Dear Anonymous Reviewer,

Thank you for the thoughtful review of my paper. Below I outline the changes made to comply with your suggestions:

  1. Abstract has been rewritten to highlight the findings of the paper.
  2. The Section 2 "Literature Review" has been added.
  3. You did not like my explanation for Equation (8). I think, that the last paragraph in the new Section 3 provides an adequate explanation. 
  4. Numbering of the equations was changed due to your suggestion. 
  5. I did not add the discussion section, but significantly enlarged the Conclusion section as to include more substantive discussion of limitations of the current approach and the implications for regulation. 
  6. Section "Funding and Acknowledgements" was added on your request. 
  7. The drawing 1C) was added to clarify the setup according to your suggestions. The caption to the figure 2 was significantly changed as to clarify the picture image. 

Please see the attachment for point to point answer.

Thank you again for the review. 

Round 2

Reviewer 1 Report

The paper was improved significantly, however, the discussion section would be much better than the extension of the conclusion section. The general comparison may always bring new insights and a better understanding of the research.  

Author Response

Dear Reviewer: 

I added a "Discussion" section at your request. It includes some material from a shortened conclusion and discussion of the regulatory challenges for high-frequency trading. 

Reviewer 2 Report

I think it can be published now, although I would have liked a better integration of the research into the current mainstream literature.

Author Response

Dear Reviewer:

In the new version, I added a discussion section, in which I outline a current state of regulation of the high-frequency trading technology. 

Reviewer 3 Report

The changes made by the author could be declared as satisfactory.

Obviously, there is an improvement in the content of the paper.

Still I ask the author for some couple of rounds proofreading and grammar check within the entire paper.

Author Response

Dear Reviewer:

In the present version, I added "Discussion" section, in which I included some material from the Conclusion, but also a discussion of a current state of regulation of the high-frequency trading technology. 

Concerning small imperfections of my English: I ran my paper through grammarly.com software for every stage of revisions. Some terms, such as "high frequency" or "high-frequency" do not have yet an established spelling.